# Which Plant Species for Green Roofs in the Mediterranean Environment?

**DOI:** 10.3390/plants12233985

**Published:** 2023-11-27

**Authors:** Luca Leotta, Stefania Toscano, Daniela Romano

**Affiliations:** 1Department of Agriculture, Food and Environment, University of Catania, 95131 Catania, Italy; luca.leotta@phd.unict.it (L.L.); dromano@unict.it (D.R.); 2Department of Veterinary Sciences, University of Messina, 98168 Messina, Italy

**Keywords:** plant species selection, ecosystem services, plant species survival, plant community, aesthetic effects, extensive green roofs

## Abstract

In recent years, owing to intense urbanization and global change with the consequent extreme climate effects, interest in green roofs, even extensive ones, in the Mediterranean environment has increased. To this end, the choice of plant species is crucial because, owing to the identification of the most suitable plants, it will be possible to expand this type of green infrastructure and increase its ecosystem services in the urban environment. In this context, the objective of the review, through a critical analysis of some of the references on the topic, is to identify suitable criteria for plant species selection that are simple to apply and able to respond to the need to have plants capable of surviving, ensuring a suitable aesthetic effect, and providing essential ecosystem services. We also investigated whether, and to what extent, associations of different species can better adapt to the difficult environmental conditions of Mediterranean green roofs. Two possible strategies to identify the plant idiotype were analyzed: the analysis of plants present in habitat analogues or the identification of morpho-functional characters capable of discriminating the response to abiotic stress, and in particular to drought stress. The use of plant communities, rather than a single species, seems capable of improving aesthetic effects, plant survival, and ecosystem services.

## 1. Introduction

The growth of the global population (in 2022, 8 billion people on Earth were exceeded) and the rate of urbanization (the UN Urbanization Statistics in 2018 [1] estimate that over 60% of the world population lived in urban areas and that by 2030, the number of megacities with more than 10 million inhabitants will be equal to 43) indicate that the problems linked to cities have become central to the quality of human life. In fact, overpopulation determines that cities have become heterotrophic organisms with their own metabolism [2], which bases their growth and expansion on the indiscriminate use of resources (energy and raw materials, often non-renewable), favored by the proliferation of means of transport and supported by industrial development and today’s technologies. Urban agglomerations return heat and pollution to the environment, alter bio-geo-chemical cycles, and cause irreversible loss and fragmentation of natural habitats [3]. Furthermore, cities not only consume the resources immediately available within their physical boundaries, but also have a pervasive effect on large areas, linked to the production of commercial goods and services necessary for their sustenance and development. To mitigate the failures of intense urbanization, a fundamental role for the environmental quality of the city is assured by green infrastructures and ecosystem services [4]. Green roofs play an important role among the green infrastructures. Since building roof surfaces cover 20–25% of urban areas [5], green roofs can effectively contribute to increasing ecosystem services (reduction in air temperature, interception of rainfall, reduction in pollution, etc.). This contribution varies among the different types of green roofs, which, based on the height of the substrate and the level of maintenance requirements, can be divided into extensive (height of the substrate often less than 10 cm, cheaper, weight approximately 60–150 kg m^−2^), semi-intensive (15–30 cm, expensive, weight 25% above or below 150 kg m^−2^), and intensive (>30 cm, very expensive, weight 200–500 kg m^−2^) [5].

Vegetation can ensure the long-term effect of a green roof, also through its evolution, as a result of interactions with the environment and between the different species used. Therefore, the identification of the most suitable plant species is one of the most important aspects of green roofs, which involves green roof designers, who must evaluate numerous parameters and individuate the advantages and disadvantages of possible plant species choices. In recent years, ecological aspects have become more prevalent than aesthetic aspects in the selection of plant species. The positive environmental effects of these types of green infrastructure in urban areas have been highlighted above all [6], which strongly influence the same criteria for choosing plants. Green roofs are considered indispensable ecosystem structures for cities, and are capable of ensuring heat island mitigation, temperature balancing of buildings, better management of water runoff, and greater urban biodiversity [7,8]. Based on the function that is considered prevalent, the selection criteria change, sometimes significantly; therefore, it is necessary to be well aware of the objectives that underlie the identification of different species [6].

## 2. The Environmental Conditions of Green Roofs

It is often thought that a green roof is a simple extension or continuation of the vegetal landscape found at ground level [9], but this is not true because the microclimatic conditions on green roofs are profoundly different; not considering this aspect can lead to high plant mortality [10]. Green roofs are difficult environments for plants, which must cope with shallow substrates, poor availability of nutrients and water, high solar radiation, and high pollutant levels [11].

Green roofs are a useful solution for covering private (terraces, roofs, and garages of residential complexes), public buildings (offices, industries, and shopping centers), or other elements that, in an urban environment, are preferred to be hidden. In all of these cases, the area explored by the root systems of the plants is strongly limited by the narrowness of the available substrate, particularly in depth. The approach used for the choice of plant species for green roofs could be compared, given the similarity of the conditions, with many situations of urban forestry, such as the same street trees in which the root systems of the plants suffer similar limitations in their development. The main consequences determined by this critical factor are different, and among the most influential:-limited growth of vegetation, determined by reduced substrate depth and rapid loss of humidity;-difficulty anchoring plants with larger sizes caused by the reduced expansion of the root systems;-inadequate drainage caused by substrate compaction.

The survival of plants on green roofs is related to different factors that modify plant physiology and growth, such as solar radiation level [12], air quality [4], and substrate characteristics [13].

While shading in most natural habitats reflects the structure of vegetation, shading in urban environments also reflects the heterogeneity in buildings and other built structures [14]. Consequently, the solar radiation reaching the plants can vary both within and between green roofs and can affect the survival of the plants as well as the temperature of the substrate. The effects of building structures can be exerted on the deposition of particulate matter [15], altering growth medium conditions and plant survivorship. Although the conditions of the green roof substrate may change over time, its composition and depth, at least in the initial phase, are designed to satisfy both the construction aspects and needs of the dominant vegetation on the green roof [16]. Therefore, differences in survival among plant species can be caused by abiotic factors associated with the design of a roof (for example, composition and depth of substrate) and by those connected to green roof sites due to the heterogeneity of the urban landscape [17].

Although natural soils are typically replaced by engineered substrates owing to their high weight, they can provide important ecological advantages for green roof systems, thereby increasing habitat vitality. Even when they cease to be biologically active, natural soils can still benefit from green roofs by mimicking the mineral properties of their natural habitat. Problems related to the infiltration of fine particles, the increased load on roofs, and the impossibility of predicting their biological activity limit their use on green roofs [18]. In addition, the compost substrate, owing to organic matter, can promote plant growth and help plants overcome the harsh conditions of a green roof [19,20]. The positive effects of deeper substrates on plant growth on extensive green roofs have been shown in various studies [19,21] and were mainly attributed to increased water-holding capacity. The characteristics of the cultivation substrates are able to promote urban biodiversity and counteract habitat loss. Schradera and Böning [22], analyzed 10 green roofs of different ages (“old roofs” were built between 1990 and 1994 and “young roofs” between 1998 and 1999) in Germany, and found that the old roofs environment is more stable due to the evolution of the soil and the greater presence of collembolans. Differences in species richness were small, but species diversity appeared to be dynamic over time, resembling that of recently reclaimed areas. A specific type of urban soil, called edifisols, is formed on buildings as a result of early and relatively natural soil formation processes occurring on artificial substrates [23]. Due to organic pollutants, plant residues, and bird droppings, a new type of humus, techno humus, is formed on these often ephemeral and young edifisols [24].

## 3. Objectives of the Plant Species Choice

The choice of plant species in relation to the type of green roof is strictly linked to the objectives you intend to achieve that vary from the simple survival of the plants to the improvement of the aesthetic effect, the reduction in costs and, above all, the achievement of particular ecosystem services.

### 3.1. Plant Survival and Cost Reduction

Plant survival on green roofs is a key parameter in the choice of plant species and clearly influences maintenance costs. Succulent plants, often characterized by the CAM photosynthetic cycle, represent the most used group of plants on green roofs [25] and are suitable for growth and survival in xeric environments, which are typical conditions of green roofs [26]. For example, the survival of forbs and grasses is typically lower than that of drought-resistant succulent plants [10], and among herbaceous plants, forbs often have a lower survival rate than grasses, at least in the natural environment [27]. Plant species with C4 photosynthesis, which are more adapted to thermal and water stress than those with C3 photosynthesis [28], can ensure greater survival on roofs in hot and dry climates. Furthermore, compared with succulents, which often form a low layer of vegetation, forbs and grasses generally have a taller stature [13], although this parameter can vary considerably between different plant species. Greater plant height and coverage of herbaceous plants could influence both the interactions between plant species mixtures and the consequent positive ecosystem functions [29]. Positive interactions can be observed between plant species [30]. For example, it has been analyzed how a greater sward height can increase the survival rate of plants on green roofs, owing to the reduction in the temperature of the soil and increased biodiversity [31]. Therefore, plant survival depends on the composition of the plant species, characterized by different growth habitus [32]; biotic and abiotic factors (e.g., light radiation, temperature, and substrate characteristics) interact to influence plant survival [17,33].

In the absence of irrigation, the selection and survival of plants depends mainly on the substrate depth of the green roof system [34]. In semi-arid regions, roofs with higher substrate depths may be required, ranging from 15 cm for succulents to over 30 cm for grasses and herbs [35]. However, a compromise must be found between substrate depth and weight because building structures often cannot support excessive weight loads [36]. The inclusion of a water retention layer, in which the water available to plants is stored for a longer period, could be a valid solution [34]. Furthermore, placing the green roof in a more sheltered position reduces the rate of evapotranspiration and helps keep the substrate moist for a longer period [12]. Solar radiation exposure influences the floral quality of green roofs in terms of vegetative cover, plant species richness, and plant species composition [37].

To identify plants capable of surviving on extensive green roofs in a Mediterranean environment, the inclusion of native plants has been proposed with positive results, even if it is not always possible to match the survival ensured by non-native species of the *Sedum* genus [38,39]. The choice of native plants, however, should not be made based only on the fact that these plants are adapted to the climatic conditions of the site [40] because the conditions in which they evolved, and above all the depth and characteristics of the soils, are very different from those of the substrate used in green roofs, which are generally shallow and well drained. Some strategies implemented by native plants, such as deepening the root system to cope with water shortages, are not applicable when there is just over 10 cm of growing substrate. Therefore, it is necessary to pay attention to plants originating from habitats characterized by shallow and well-drained substrates, such as rocky or ruderal habitats; in the communities that settle in these contexts, there may be plant species suitable for extensive green roofs, in particular.

The use of native plant species on green roofs, especially in the Mediterranean area, is also limited by the lack of references regarding their performance and necessary cultivation care. Furthermore, the seeds of native plant species do not simply bud on rooftops [41].

The identification of plants with characteristics and physiology suitable for the toleration of stress can significantly reduce irrigation needs and related maintenance costs, a particularly important aspect in a climate, such as the Mediterranean climate where water is scarce in summer [42].

### 3.2. Aesthetic Effects

To achieve interesting ornamental effects, it is best to aim for the presence of a plant community rather than a single species. *Sedum* monocultures, for example, are widely adopted in extensive greenery, and have a monotonous effect [43], which is not always pleasant. Furthermore, similar to agricultural monocultures, the presence of a single plant species increases the risk of parasitic attack. Abscission of leaves in response to drought, typical of many native Mediterranean species, can improve the response to water stress [44]; however, from an aesthetic perspective, it is an aesthetically unpleasant adaptation.

Obtaining a space of high aesthetic value is an important ecosystem benefit provided by green roofs [45], although this is not always recognized. The contribution of green roofs to stress reduction in modern cities was analyzed (through the psychological and physiological parameters of heart rate variability and systolic blood pressure). Based on these results, it was observed that a space with an open and structured vegetation design that includes both grass and shrubs may have more potential for stress reduction than a monotonous vegetation model; the increase in vegetation biomass was not necessarily linked to higher psychological and physiological benefits [46].

Several studies have suggested that future research on plant breeding should better consider and quantify plant architecture and shape, flowering duration, variation in color and vegetation throughout the year, biomass production, and how irrigation and other cultivation techniques influence the vegetation and aesthetic performance of green roofs [36,47].

### 3.3. Ecosystem Services

There are numerous ecosystem services provided by green roofs. Among the main and typical ones, there is rainfall interception and prevention of flash floods in urban areas, consequently decreasing the load on sewer systems. To this end, vegetation plays an important role because it significantly influences the moisture content of the substrate and runoff rate [48,49]. The reduction in runoff occurs through various processes such as water interception, plant transpiration, root absorption, retention, and storage of water in plant tissues [50]. Plant water consumption is determined by transpiration capacity, whereas root biomass influences its ability to store water [51]. Increasing the plant cover on a green roof improves rainwater retention [52]; however, species richness does not significantly affect water storage unless diverse plants are characterized by higher water consumption [53]. Beyond the choice of plant species, there is no doubt, however, that an element that influences the interception of water is the depth of the substrate: Soulis et al. [54], using three species with different characteristics—*Sedum sediforme* (Jacq.) Pau, a succulent plant, *Origanum onites* L., a xerophyte plant, and *Festuca arundinacea* Schreb., a turfgrass—noticed how the total runoff reduction (%) for the entire study period was equal for the three species to 50.8%, 63.6%, and 54.9% with a substrate depth of 8 cm, and to 60.3%, 81.1%, and 68.8%, respectively when the substrate was 16 cm deep. However, it should be emphasized that at the same depth of substrate, xerophyte species were the most efficient in intercepting rainwater.

The amount of runoff can be reduced by increasing the number of plant species with different characteristics [55]. The presence of taller species can lead to a higher rate of water interception and evapotranspiration, whereas the presence of shorter plants in contact with soil determines the presence of a layer of humidity beneath the canopy. The capacity to store water changes based on the characteristics of the plant (orientation and size of the leaves, roughness and hydrophobicity of the bark, rigidity of the branches, and density of the foliage). Plant community structure can also influence water interception [51]. To this end, the characteristics of the root system are important because water retention in the substrate is influenced by the root structure. Plants that form dense and fibrous roots, such as grasses [51], reduce the porosity of the substrate and the volume available to retain water [56], thus proving to be more efficient in reducing rainwater runoff than succulent or forb species. Evapotranspiration is the other plant-related process that influences runoff production [57]. Species with high evapotranspiration rates that reduce water from the growing medium create more space for the capture of water in subsequent rain events [56].

Another ecosystem function of green roofs is the increase in biodiversity; green roofs can also be used as an urban space where plant biodiversity can be preserved [58]. Higher plant species biodiversity can make extensive green roofs attractive to native arthropods and avian species [59]. Having different plant species, especially in extensive green roofs, may allow them to adapt to variable moisture conditions and maximize the evaporative cooling benefit, thus extending the benefits of green roofs [60].

Plants absorb a significant percentage of solar radiation through their biological functions such as photosynthesis, transpiration, respiration, and evaporation [61]. Similarly, it has been suggested that the presence of different plant species with lower reflectance to increase diversity would only reduce the albedo of green roof [29]. Therefore, the choice of plant species has an important effect on temperature [62]. It has been suggested that plant characteristics, such as height, leaf area index, and plant responses to drought, greatly influence the thermal insulation capacity of green roofs [63]. Plants with a high canopy density and height can provide greater shading and cooling via transpiration, suggesting that plant phenology, plant water consumption, and their interactions are key points for understanding seasonal differences in the relative contribution of each plant, especially for roof thermal performance [62].

The greater thermal regulation capacity of vegetated roofs compared with non-vegetated roofs has been largely attributed to the shading effect caused by plants or greater evapotranspiration [64]. Since these two factors are not always directly related, the selection of plant species and adoption of models to facilitate this selection are complex. In this regard, Azenas et al. [65] highlighted how *Sedum sediforme* (Jacq.) Pau, a species that consumes little water and forms a dense, multi-layered foliage of fleshy leaves, has a higher thermal regulation capacity than *Brachypodium phoenicoides* (L.) Roem. & Schult., a herbaceous species with higher transpiration rates than *Sedum* sp. This suggests that canopy architecture, leaf anatomy, and spatial arrangement of leaves would have a greater effect on thermal regulation than the transpiration capacity of plants. Although both species show similar surface coverage, *B. phoenicoides* has erect leaves that would allow for the passage of a greater quantity of incident radiation compared to *S. sediforme*, a species that presents creeping behavior, with maximum development and flowering during the warm season. If shading has positive effects on summer thermal regulation, it has negative effects in winter because it reduces the temperature. Nevertheless, the performance of the plants must always be considered based on the climatic conditions of the site. In fact, some models have indicated that taller plants have a greater insulating effect in Mediterranean European cities due to the height of the foliage and higher LAI values [66]. However, these models usually do not consider the differences in leaf anatomy and canopy architecture, which affect the thermal regulation capacity of plants.

Blanusa et al. [64] analyzed different types of plants (a mixture of *Sedum*, *Stachys byzantina* K.Koch, *Bergenia cordifolia* Sternb., and *Hedera hibernica* Carrière) to evaluate whether and to what extent the plants differ in their “cooling potential”. They wanted to understand whether leaf morphology influenced leaf temperature, and how the substrate water content altered this response. The relationship between the temperatures of the leaf surface and those of the air immediately above the canopy and the influence of the type of plant on the temperature of the substrate below the canopy (i.e., potential to provide aerial cooling) were also studied. It was found that *S. byzantina* offered the best results in terms of cooling the leaf surface (even in the drying substrate, e.g., 5 °C cooler than *Sedum*), substrate cooling under the canopy (up to 12 °C), and air above the canopy (up to 1 °C, when soil moisture was not limited). Based on these results and the significance of the reduction in temperature, the authors suggested that the choice of plant species on extensive green roofs should not be anchored only to the survival of the plants, but should also consider the contribution in terms of ecosystem services. Plant selection should therefore be based not only on “what survives”, but also on “what provides the best ecosystem service” [64].

The effectiveness of the temperature reduction is also a function of the height of the plants, and the highest temperature reductions were obtained with plants 35 cm tall, followed by those 15 cm tall, and then by those 10 cm tall. Plants with green leaves are more efficient at reducing temperatures than those with purple/red leaves [67].

Another ecosystem service is the ability of the green roof to control and reduce sound pressure [68]; the reduction in the noise level can reach 8–10 dB [5] in relation to the vegetation layer characteristics.

Finally, we cannot forget that green roofs can be used as supplements for other sources of food production in urban areas. Urban agriculture carried out on roofs can improve numerous ecosystem services, increase biodiversity, and reduce food insecurity [69].

## 4. Criteria for Plant Species Selection

The selection criteria, or rather the idiotype of plant species or plant community, change for extensive or intensive green roof; for the first, plant species that tolerate water deficit in the substrate are recommended, with a low and prostrate habit (to obtain good coverage), evergreen foliage, and a long flowering season. However, in intensive green roofs, it is possible to use a wide range of plants, including shrubs and trees, thus increasing the ability of plant cover to reduce pollution and improve air quality [70]. It must be taken into account that fertilization, carried out in an intensive green roof to promote plant growth, makes plants more vulnerable to drought [71]. The use of numerous plant species to improve biodiversity favors vegetation establishment [57].

The selection of plants must consider various aspects (Table 1), such as weight tolerated by the structure, plant cover, level of maintenance, and tolerance to the difficult environmental conditions that occur in green roofs, such as high exposure to the sun, shallow growing substrates, limited water availability, increased wind speed, and prolonged drought periods [72].

Species of the *Sedum* genus, which are succulent groundcovers, are widely used in green roofs because they can survive in difficult environments [73]; however, some researchers suggest favoring native plant species due to their high adaptability to local climatic conditions [74]. To expand the list of plants suitable for green roofs, it is possible to include species that live in habitats similar to those found on roofs, such as coastal dunes and rocky outcrops exposed to high temperatures and shallow soils [75]. According to Balachowski and Volaire [76], it is good to consider both the climate of origin and the characteristics of the plant when choosing the species, as the former indicates the environment in which a plant is adapted, but the latter allows for predicting the probable ecological strategies that the plant will use in response to resource limitation [77].

There are different approaches to selecting plants suitable for green roof environments: they can be based on the characteristics of the plants, through a morpho-physiological and ecological approach [51,78,79], or on the identification of natural habitats that express conditions similar to those of green roofs [75,80]. Some physiological approaches have tested the ability of plants to tolerate drought and utilize excess water when present, as these strategies are functional for selecting plant species for green roofs where stormwater management is relevant [81]. Another strategy is based on climatic characteristics of the area of origin. Climate is the key determinant of large-scale ecosystem composition and structure, and temperature and precipitation significantly influence plant species distribution [82]. Aridity indices, such as the heat moisture index (HMI) [83], can be useful because they represent the aridity of the environment and consider the interaction between temperature and precipitation [84]. Plants from climates with a higher HMI tolerate drier and hotter conditions, which may indicate greater tolerance to water deficit conditions [85]. Shrubs can also be very drought tolerant, for example, those from dry rocky habitats have been found to survive water-deprived conditions when placed in a green roof with a shallow substrate [81]. Although several studies have analyzed the drought tolerance of shrubs in arid and semi-arid habitats [63,86,87], no studies have evaluated shrubs based on physiological approaches and climate of origin. Regarding stormwater retention, ideal shrub species for green roofs should have high evapotranspiration under well-irrigated conditions, but tolerate water deficit conditions, as attested by a more negative midday water potential [81]. This strategy has been demonstrated for monocotyledonous and herbaceous species from rocky outcrops, which showed high water use under well-irrigated conditions and drought tolerance by reducing water potential under water-deficit conditions. However, it is unclear whether this combination of high water use and high drought tolerance also exists in shrubs because drought tolerance is often negatively correlated with transpiration rate [88].

### 4.1. The Idiotype of Plant Species

Several methods have been proposed to select plant species on a green roof, such as the “habitat template” approach proposed by Lundholm and Walker [80]. This approach is based on the concept that artificial ecosystems can be modified to present conditions similar to those of a natural habitat so that plants living in that environment are able to adapt better to green roofs [57]. However, another approach comprises selecting a certain characteristic of the plant that appears strategic [79,89,90], such as the size of the plant, aesthetic value [36], or resistance to abiotic stress [91]. The key assumption of the latter approach is that abiotic stresses are critical factors that limit plant growth and survival, particularly in green roofs.

Another factor that influences the plant selection process is canopy structure. Plants with a mostly horizontal leaf distribution and/or extensive foliage development should be selected to reduce the transmission of solar radiation. Other important factors include growth rate, nutrient requirements, and sensitivity to pollution. In addition to drought, plants on green roofs must tolerate intense radiation and high temperatures in both air and substrate. In particular, high thermal levels in the root zone can have detrimental effects on plant physiology, growth, and biomass [92]. Savi et al. [87] reported a significant positive correlation between the root vulnerability to heat stress and plant mortality. Ideal plants should be drought, heat and wind tolerant, pest resistant, light loving, have low height, good coverage, and shallow root system [93].

It is advisable to favor plants that express the following characteristics: ease of transplant, simple maintenance, slow growth rate, resistance to pollution, and capability to absorb or retain pollutants. Plants must be able to survive even in the presence of large quantities of water, because there is no good drainage system on some roofs [93]. Being taller, trees are more exposed to the action of wind than other types of plants; the size of the canopy and leaves influences their susceptibility to wind; therefore, it is often necessary to provide anchoring [94].

Ideal plants for extensive green roofs must be established quickly, provide high ground cover, tolerate extreme environmental conditions, and adapt to limited substrate depths [38]. Therefore, it is not surprising that succulents are among the most intensively studied taxa, as they have shallow root systems, high water use efficiency, and are able to tolerate the extreme conditions found on extensive green roofs in Northern regions with growth substrates of 7–10 cm, but also 5 cm. The *Sedum* genus has proven to be very reliable, as it has high drought tolerance. Many species tolerate up to one month without precipitation and some *Sedum* species have survived for up to four months without water in greenhouse experiments in the Great Lakes region of the United States [95]. The genus *Delosperma* also appeared to be suitable and *D. cooperi* (Hook.f.) L.Bolus and *D. nubigenum* (Schltr.) L.Bolus proved to be suitable in many regions [96]. Some perennial grasses and herbaceous flowers also have effective adaptive measures to cope with drought. The answer depends on climatic conditions and substrate depth; with substrates of 15 cm or more, many native herbaceous plants survive in northern climates without irrigation [97].

Geophytes can survive periods of drought thanks to the accumulation of water in underground organs, such as bulbs and rhizomes [98]. Especially in a Mediterranean climate, therophytes appear promising because their short life cycle allows them to spend the summer months dry as seeds and to germinate again after autumn rain [89].

Green roofs are often installed to reduce urban stormwater runoff; to achieve this, plants must use water when it is available, but reduce transpiration when water itself is in limited supply. Succulent species often fail to achieve these goals. To this end, Farrel et al. [81] observed the behavior of species found in granite outcrop environments to evaluate the water use strategies they implemented under contrasting conditions of water availability. The investigated species exhibited good plasticity with respect to water use. The authors developed a conceptual model using physiological traits to select species suitable for green roofs. Ideal plant species are those that use high quantities of water and can tolerate drought simultaneously. In particular, the parameters analyzed were (i) water use strategy under well-watered conditions (WW), (ii) status rate of water use under water-deficit (WD), and (iii) maintenance of water under WD conditions. Suitable plant species are those with high use of water under conditions of good water availability, which are therefore able to reduce the runoff of rainwater and maintain a good water status under conditions of water deficit. Species with slow rates of water use under water deficit conditions are more tolerant to drought.

In the study, carried out in Australia, in contrast to the succulent (*Sedum pachyphyllum* Rose), used as a control, all the species of the granite outcrop showed plasticity in the use of water; however, the most drought-resistant species proved to be the four monocotyledons (*Arthropodium milleflorum* (Redouté) J.F.Macbr., a geophyte, *Stypandra glauca* R.Br., *Dianella admixta* Gand., *Lomandra longifolia* Labill., hemicryptophyte grasses) and the flowering herbaceous plant, also hemicryptophyte, *Isotoma axillaris* Lindl., present in the surface depressions on the granite outcrops. This was achieved through reduced transpiration and/or a high root mass fraction (RMF), proving that high water users can also be drought resistant, making them the most suitable plant species for green roofs. All species analyzed had a certain degree of root, stem, or leaf succulence. Therefore, plant selection for green roofs should prioritize these traits in non-succulent species. Shrubs, such as *Grevillea alpina* Lindl. or *Correa reflexa* Vent., present in these habitats, are generally found in deeper depressions in the soil or in cracks [99] and can avoid water stress by accessing water stored deeper in the underlying rocky substrate [100]. This strategy, which is useful for granite outcrops, is not effective for green roofs with shallow substrates; therefore, the high water potential and anisohydric response of shrubs to water deficit is a risky strategy [81]. Other selection criteria, such as plant shape and aesthetic value, can be considered, although an approach based on the physiological response to water availability also has great potential to improve the ecosystem services provided by plants [81].

Lundholm et al. [78] used four characteristics (height, individual leaf area, specific leaf area (SLA), and leaf dry matter content) to evaluate whether these characteristics could predict adaptability to green roof conditions and ecosystem services. Six indicators were considered for the latter: thermal, hydrological, water quality, and carbon sequestration functions. Species average height and SLA were the most useful traits for predicting several services via their effects on canopy density or growth rate. The characteristics of plants, which are easily measurable, can therefore be used to select species suitable for optimizing the performance of green roofs, as well as from an ecosystem service perspective [78].

### 4.2. Choose the Plant Species or the Plant Community?

Although it might be tempting to use only the optimal plant species for green roofs, it is better to use plants of different biological forms and, therefore, different drought tolerance strategies to ensure both better performance and resilience to stress [101].

The performance of green roofs, in fact, could be improved by expanding the range of plant species currently used [32,47,102]. Plant diversity, in terms of both life form and drought tolerance, can improve ecosystem functioning, due to the complementarity of performance [32,37]. Numerous studies underline the importance of biological interactions between different plant species that influence the survival and growth of plants; therefore, it is preferable, when choosing the plant species, to refer to the entire plant community rather than to a single plant species [103]. Species composition of plant communities is determined by their interactions with biotic and abiotic factors. Historically, abiotic factors (soil moisture, temperature, nutrients, and pH) were thought to have a more significant influence on the establishment and survival of plant species, and the role of biotic interactions, whether negative (competition) or positive (facilitation), was underestimated in plant development and in the coexistence and structure of plant communities [104,105].

*Sedum* species can facilitate the survival of non-succulent plants by reducing high soil temperatures during drought periods [102]. Their particular photosynthetic pathway (CAM) and succulent leaves allow them to survive prolonged periods of drought, making them the main choice for extensive green roofs [95]. Even bryophytes, which can spontaneously colonize substrates and survive extreme droughts, can mitigate the thermal conditions of extensive green roofs, thereby facilitating the survival of other vascular plants [102].

The presence or absence of these interactions influences relationships within and between plant communities [106]. In particular, in the context of green roofs, great interest has been aroused by the so-called “nurse plants” which facilitate the growth of other plants that live within their canopy, modifying the microclimate [107], providing shade in conditions of high temperatures [102], increasing soil nutrients with an increase in decomposing organic matter [108], reducing the negative action of the wind [109] or increasing the soil moisture content [110]. Furthermore, the presence of nurse plants increases the biodiversity of species, abundance of plant formation, and survival of other plants [11]. Butler and Orias [102] analyzed the use of *Sedum album* L. as a nurse plant to facilitate the growth of *Agastache rupestris* (Greene) Standl. and *Asclepias verticillata* L. and noted how *S. album* acted as a competitor during periods of abundant water, reducing the biomass of the other two species, while improving their performance during the summer water deficit, and reducing their mortality.

When plants with different habitus are adopted (succulents, forbs, and grasses), better management of rainwater and greater cooling are achieved compared to monocultural systems [34]. One of the most important interaction effects between plants is connected to the shading exerted by taller individuals on shorter ones. In the absence of this protective action, high solar radiation can reduce photosynthetic activity due to the photoinhibition of PSII [111], with consequent delay in plant growth [112], whereas the high leaf temperatures that are reached lead to the closure of the stomata and inactivation of the enzymes, with a consequent reduction in growth [113]. In particular, in hot climates, shade is observed to increase the growth of many plant species, owing to the involvement of microclimatic conditions [11]. The level of shade tolerable by plants is affected by their light compensation point (LCP), and the LCP of different species can be adopted for the selection of ornamental plants that can be used in urban and peri-urban areas [114] and green roofs. Furthermore, in green roofs, along with the shading caused by vegetation, the shading cones of the neighboring structures are of significant importance and can significantly modify the microclimatic conditions.

Linking plant water use to leaf traits and Grime’s CSR strategies [104] could facilitate the selection of green roof plant. To this end, Lönnqvist et al. [115] determined the growth (shoot biomass, relative growth rate, and leaf area), leaf characteristics (leaf dry matter content, specific leaf area, and succulence), and CSR strategies of 10 common species on European green roofs [of which three succulents: *Phedimus spurium* (M.Bieb.) ‘t Hart, *Hylotelephium telephium* (L.) H.Ohba, *Sedum acre* L.; one grass: *Poa alpina* L., and six forbs: *Campanula rotundifolia* L., *Galium verum* L., *Hypericum perforatum* L., *Lotus corniculatus* L., *Tanacetum vulgare* L., and *Viola tricolor* L.], and correlated them with their use of water under conditions of full availability (WW) or water deficit (WD). All three succulent species included in the experiment mostly showed stress tolerance traits, and their water loss was lower than that of the unvegetated substrate, probably due to substrate cover. Plants with higher water use under WW had ruderal and competitive strategies, and greater leaf area and shoot biomass than those with lower water use under WW. However, the four species with the highest water use under WW conditions (*T. vulgare*, *V. tricolor*, *P. alpina*, and *L. corniculatus*) were able to reduce their water use under WD, indicating that they could both retain more precipitation than surviving periods of water limitation. This study indicates that for optimal rainwater retention, the selection of green roof plants in the regions, at least in the northern European latitudes, should focus on the selection of non-succulent plants with predominantly competitive or ruderal strategies to maximize the long day length during the short growing season.

The diversity of plant species and functional traits have been shown to enhance green roof ecosystem services. Differences between plant species that contribute to ecosystem services are products of evolutionary change and phylogenetic diversity (PD). MacIvor et al. [116] analyzed six combinations of communities with different levels of PD, using 28 plant species from 12 different botanical families. They found that the minimum and average roof temperatures in the plant community decreased with an increase in PD. The increase in PD also led to an increase in the volume of rainwater captured, although it was not proportional to the amount of water lost due to evapotranspiration 48 h following the rainfall event [116].

Nagase and Dunnet [47] investigated the influence of biological diversity on the survival of plants subjected to water stress. Twelve plant species were selected from the three main functional groups commonly used for extensive green roofs (*Sedum*, forbs, and grasses). For each group, four species were selected and planted in combination with increasing diversity and complexity: monocultures, four-species mixtures, and twelve-species mixtures. Three irrigation regimes were imposed: watering every one, two, or three weeks. The results demonstrated that a more diversified mixture was more advantageous than a monoculture in terms of survival and presence of vegetation. Combinations of functionally diverse species can achieve this goal more effectively than plants of the same taxonomic group, which, compete for the same resources when grown together [47].

If a single function improves in a more plant species-rich community than in a monoculture, this is called a “mixture advantage” which is based on two factors: the first is connected to the benefits associated with the cultivation of different species; the contribution of a plant species to a certain function improves when the species itself is cultivated with another species [117]; the second is niche complementarity [117] and occurs when two or more plant species, due to differences in resource use, can better exploit available resources. Therefore, it is important for plant species to exhibit functional differences. In constructed ecosystems, designers are unlikely to rely on chances to improve ecosystem function; therefore, it is important to identify the most important functional traits that determine ecosystem functioning and to determine whether species combinations can reliably improve ecosystem function. If the plant species in a mixture are relatively similar, for example, all succulent plants, functional diversity will be low despite the high taxonomic variability.

## 5. Plant Species for Green Roofs in the Mediterranean Area

In many regions with hot and dry climates, including the Mediterranean, green roof technology is not widespread [35], mainly due to the difficult climate (summer drought and high temperatures) and the limited availability of water. These characteristics impose severe restrictions on the growth and survival of green roofs [36,102]. Plants are assumed to not survive in semi-arid climates on unirrigated green roofs with substrate depths of less than 5 cm, especially during the summer drought or establishment phases [35,118]. Furthermore, summer water scarcity is a recurring problem in the Mediterranean and climate change will lead to even more severe water scarcity because summer precipitation is expected to decrease by 5% per decade [119]. This can lead to irrigation becoming an unsustainable, regulated, and limited option. Therefore, it is necessary to select plant species that are capable of adapting to the absence of irrigation [89]. Mediterranean areas contain habitats rich in native plant species that have the potential to be used on extensive green roofs [89] because they are believed to be better adapted to local climatic conditions and require little maintenance [34]. From a biodiversity perspective, one could also assume that green roofs constitute a new habitat for some Mediterranean plants whose natural environments are in danger. The choice can rely on the fact that many native plant species of the Mediterranean (in particular, the xerophytes) express morpho-functional and physiological adaptations that make them particularly suitable for green roofs; in fact, they present changes in the structures of the leaves (imbricated or often linear, with thick and waxy cuticle, sunken stomata, pubescent surface) and roots (deep roots, large root hair, rapid development in young plants), reduction in photosynthesis, and leaf drop phenomena under conditions of drought, high solar radiation, and high temperatures in summer [42].

Although research in the Mediterranean environment is less extensive than that carried out in a continental climate, numerous native species have been considered. Azenas et al. [120], for example, analyzed the response of five Mediterranean species—*Brachypodium phoenicoides* (L.) Roem. & Schult., *Crithmum maritimum* L., *Limonium virgatum* (Willd.) Fourr., *Sedum sediforme* (Jacq.) Pau, and *Sporobolus pungens* Kunth—grown in a non-limiting water regime or restoring 50% of evapotranspiration. The plant species were selected because they grow in natural habitats characterized by shallow soils with low organic substance content, high solar radiation, and extreme temperatures, namely, under conditions similar to those found on a green roof. Furthermore, some of these (*S. sediforme*) were CAM or CAM-facultative succulent species. The behavior of these plant species was monitored for two years. All plant species survived and exhibited a suitable aesthetic performance and vegetation coverage. *S. sediforme* recorded the least changes in appearance, the highest biomass production, and the lowest water consumption. However, *B. phoenicoides* appears to be an interesting alternative due to its valuable aesthetic characteristics and water consumption during the rainy season, suggesting a potential role of this species in the regulation of rainwater related to runoff reduction. *S. pungens* performed well in summer but presented poor aesthetic value during winter. *L. virgatum*, a plant that grows on rocky coasts, has shown good aesthetic value, both for its flowering and compact shape, and a high carbon sequestration capacity. In contrast, the use of C4 species, such as *S. pungens*, in urban green roofs in the Mediterranean climate is limited by the difficulty of this species to survive in winter and regrow in early spring [120].

To broaden the diffusion of green roofs in the Mediterranean environment, the contribution of four native plant species in Portugal—*Antirrhinum linkianum* Boiss. & Reut., *Asphodelus fistulosus* L., *Centranthus ruber* (L.) DC. and *Sedum sediforme* (Jacq.) Pau—resilient and drought tolerant, was analyzed. Growth, and aesthetic value were evaluated under two irrigation regimes (return of 100 and 60% evapotranspiration). *A. linkianum* had the highest number of flowers, longest seed production duration, and the highest area coverage, demonstrating its suitability for use. The level of irrigation did not significantly affect flowering and green coverage for any of the plant species and irrigation costs could be reduced by adopting deficit irrigation [121].

Attention has also been paid to therophyte species; annual plants contribute significantly to the vegetation of the Mediterranean basin, but their presence on green roofs has been limited to date [89], which is due to the brevity of their cycle, the difficulty of regeneration, and the lack of competitiveness compared to perennial plants. The absence of these plants during the summer months results in a modest cooling effect during the hot season. Van Mechelen et al. [89] analyzed the plants present in natural habitats in southern France, that presented characteristics similar to those of green roofs, and identified 372 potentially usable species on the basis of some functional parameters; of these 35% are therophytes, which indicates that many annual species can be taken into consideration. Therophytes have interesting properties such as a short flowering period and the production of many seeds. Their conservation value may also be important, as many annual plants are threatened in Mediterranean areas [122]. Other traits such as CAM metabolism, stress tolerance, and succulence have been shown to be important for successful green roofs [25].

The screening tool provides a potential list; however, definitive proof is obtained through experimental tests. Despite these limitations, the plant characteristics approach offers interesting possibilities for Mediterranean regions and can also help adapt green roof designs to future climate change [89].

Sage species native to Greece—i.e., *Salvia fruticosa* Mill., *S. officinalis* L., *S. pomifera* ssp. *pomifera*, *S. ringens* Sm., *S. tomentosa* Mill.*,* and interspecific hybrids—were evaluated for their inclusion in an extensive green roof in a Mediterranean climate in the summer period with regular or reduced irrigation (every 2–3 days with substrate humidity of 16–22% *v*/*v* and 4–5 days with substrate humidity of 7–11% *v*/*v*). Regardless of the irrigation frequency, *S. pomifera* ssp. *pomifera* x *S. ringens* and *S. officinalis* x *S. pomifera* ssp. *pomifera* showed the highest survival rate among all hybrids and species, as well as satisfactory growth, while S*. fruticosa* recorded the lowest survival, demonstrating that numerous *Salvia* species can be used in extensive green roofing in arid regions [123].

The possible use of two Mediterranean shrubs, *Arbutus unedo* L. and *Salvia officinalis* L., on green roofs was analyzed. The first species presented a substantial isohydric response (owing to the reduction in the stomatal opening at the first signs of stress, it was able to contain an excessive lowering of the water potential) and the second anisohydric (the plant appeared capable of withstanding strong variations in water potential while only partially limiting stomatal closure). Both species can be used on the Mediterranean green roof, even if the anisohydric species appear to be more sensitive to the characteristics of the substrate [63].

Reducing soil temperature while maintaining a relatively high air temperature has been shown to improve the growth and functional status of both roots and shoots and enhance plant survival [124]. This contrasts with the need to reduce the depth of the substrate to limit weight and installation costs [125]. However, the depth of the substrate is not always a limiting factor in the adoption of shrubs. Savi et al. [126] investigated the behavior of two drought-adapted shrubs of two-years-old (*Cotinus coggygria* Scop. and *Prunus mahaleb* L.) grown in experimental modules with a 10 or 13 cm deep substrate. The results highlighted how the reduced depth of the substrate translated into less severe water stress than hypothesized and that the shallower substrate indirectly stimulates lower water consumption as a consequence of the reduced plant biomass; therefore, it is possible to hypothesize a green roof with the use of stress-resistant shrubs in sub-Mediterranean areas, even in the presence of a substrate only 10 cm deep.

The performance of native Mediterranean plants on green roofs could be improved by adopting a plant community instead of a monoculture. Varela-Stasinopoulou et al. [127] analyzed the growth, flowering, and self-reproduction rate of three plant communities, artificially created and made up of native Mediterranean plants, placed in substrates of different depths (8 and 15 cm) and with two irrigation regimes (high, 20% ETo and low, 10% ETo). The plant communities simulated those on the islands of Crete and Greece. Each of the three artificial plant communities comprised nine species and subspecies. Deeper substrates significantly improve the growth, flowering, and survival of most taxa. The irrigation regime was not significant for any species except for one, indicating that minimal amounts of water may be sufficient for irrigation. Four species failed to flower, whereas 15 species managed to self-reproduce.

Information on the plant species proposed for extensive green roofs in the Mediterranean region is presented in Table 2. The selected papers were experimental trials conducted in the Mediterranean environment. No information has been reported on plant species performance because the operative and stressful conditions are quite different. The list is full of over 180 species and/or cultivars belonging to over 40 families, most of them of Mediterranean origin, attesting to a large number of plant species that can be counted even with substrate depths of less than 20 cm. Among biological forms, chamaephytes (~40% of the total) and hemicryptophytes (~30%) stand out.

## 6. Conclusions

Green roofs are increasingly being considered for their contribution to ecosystem services, which are essential for the quality of life of urban areas. Many of its functions are connected to the vegetation that is installed, making the choice of species one of the most strategic aspects of the design. Even in the Mediterranean environment, sustainable solutions are possible for the creation of a green roof, which must be based on a reduction in the height of the substrate to allow it to be installed in a greater number of structures, and a more rational use of water resources. For the identification of species, the analysis of both plant species is present in habitat analogs and some morpho-functional characteristics can be considered a useful strategy. However, only experimental trials in representative contexts can provide conclusive information to understand whether these choices are functional. Another strategy is to increase biodiversity by replacing a single species with a plant community to make the best use of the positive relationships that can be established between different plant species. However, even in this case, only specific experimental trials would be able to provide a framework of knowledge indispensable for spreading the presence of green roofs in the Mediterranean environment.

## Figures and Tables

**Table 1 plants-12-03985-t001:** Key traits of plant species suitable for green roofs in Mediterranean environment.

Ecological	Ecosystem Service	Morpho-Biometric	Aesthetic	Physiological	Cultivation
- plant diversity and symbiosis; - native origin; - competitive or ruderal Grime’s strategies.	- good rainfall interception; - good dust capturing ability; - resistant to pollution and capable of absorbing or retaining pollutants; - reduction in noise level; - thermal regulation; - increase biodiversity.	- low plant height with cushion forming habit; - shallow and spreading roots; - root system adapts to limited substrate depth; - leaf shape and orientation; - high LAI value; - structure wind tolerant; - low shrubs, forbs, grass, groundcover plants, geophytes, and climbing plants.	- plant shape and aesthetic appearance; - relatively stable growth rate and high ornamental value; - high ground cover; - establish quickly; - long flowering season; - variation in color and vegetation throughout the year.	- succulent leaves with ability to store water; - certain degree of succulence of roots and stems; - flooding tolerance; - tolerance to high/load temperature; - drought-tolerance and ability to utilize excess water; - tolerate extreme environmental conditions; - CAM or C4 photosynthesis; - ability to adapt the transpiration rate to environmental conditions; - plasticity in the use of water; - light-loving plants.	- easy to transplant; - short period of establishment and fast reproduction; - have been used locally and commonly; - have been successfully introduced; - simple maintenance and with a slow growth rate, and low nutrient requirements.

**Table 2 plants-12-03985-t002:** Plant species studied for extensive green roofs ^1^ in the Mediterranean region.

Plant Species ^2^	Botanical Family ^3^	Plant Life Form ^4^	Chorotypes ^5^	Substrate Depth (cm)	References
*Achillea millefolium* L.	Asteraceae	H	Eurosib.	4, 7, 10, 19	[77,128]
*Aeonium arboreum* Webb & Berthel.	Crassulaceae	NP	Macarones.	6	[129]
*Allium carinatum* L.	Amaryllidaceae	G	Stenomedit.	20	[130]
*Allium roseum* L.	Amaryllidaceae	G	Stenomedit.	10	[131]
*Allium sphaerocephalon* L.	Amaryllidaceae	G	Paleotemp.	5, 10	[37]
*Alyssum alyssoides* L.	Brassicaceae	T	Eurimedit.	5, 10	[37,131]
*Alyssum saxatile* L.	Brassicaceae	C	Medit.	14	[132]
*Anemone hortensis* L.	Ranunculaceae	G	N-Eurimedit.	20	[130]
*Anthemis arvensis* L.	Asteraceae	T	Stenomedit.	20	[130]
*Anthemis maritima* L.	Asteraceae	H	W-Medit.	15, 20	[36]
*Anthyllis vulneraria* L.	Fabaceae	H	Eurimedit.	10	[131]
*Antirrhinum majus* L.	Plantaginaceae	C	W-Medit.	20	[130]
*Antirrhinum linkianum* Boiss. & Reut	Plantaginaceae	C	Endem. Portugal	11	[121]
*Aptenia cordifolia* (L.f.) Schwantes	Aizoaceae	C	Africa	5, 6	[129,133,134]
*Arbutus unedo* L.	Ericaceae	P	Stenomedit.	18	[63]
*Armeria maritima* (Mill.) Willd.	Plumbaginaceae	H	Subcosmopol.	4, 7, 10, 11 ± 1	[128,135]
*Armeria maritima* (Miller) Willdenow subsp. *maritima*	Plumbaginaceae	H	Subcosmopol.	19	[77]
*Armeria maritima* ‘Rosea’	Plumbaginaceae	H	Subcosmopol.	14	[132]
*Armeria pungens* Hoffmanns. & Link	Plumbaginaceae	C	W-Europ.	15, 20	[36]
*Artemisia absinthium* L.	Asteraceae	C	E-Medit.	7.5, 10, 15	[19,86]
*Arthrocnemum macrostachyum* (Moric.) K.Koch	Amaranthaceae	C	Medit.	10	[136,137]
*Asphodelus fistulosus* L.	Asphodelaceae	H	Subtrop.	11	[121]
*Atriplex halimus* L.	Amaranthaceae	P	Stenomedit.	7.5, 10, 15	[19,138]
*Atriplex portulacoides* L.	Amaranthaceae	C	Circumbor.	10	[139]
*Ballota acetabulosa* Benth.	Lamiaceae	C	W-Asia	8	[140]
*Blackstonia perfoliata* (L.) Huds.	Gentianaceae	T	Eurimedit.	10	[131]
*Brachypodium phoenicoides* (L.) Roem. & Schult.	Poaceae	H	W.Stenomedit.	15	[42,65,120,141]
*Brachyscome multifida* DC.	Asteraceae	H	Australia	19	[77]
*Calamintha nepeta* Savi	Lamiaceae	C	Medit.-Mont.	15, 20	[36,130]
*Calendula arvensis* L.	Asteraceae	H	Eurimedit.	10	[131]
*Carpobrotus edulis* (L.) N.E.Br.	Aizoaceae	C	S-Africa	5, 6, 9, 12, 15	[133,142]
*Carpobrotus rossii* (Haw.) Schwantes	Aizoaceae	C	S-Africa	10	[143]
*Carthamus carduncellus* L.	Asteraceae	H	NW-Medit.	5, 10	[37]
*Centaurea cyanus* L.	Asteraceae	T	Stenomedit.	20	[130]
*Centranthus macrosiphon* Boiss.	Caprifoliaceae	H	W.Stenomedit.	10	[131]
*Centranthus ruber* (L.) DC.	Caprifoliaceae	C	Stenomedit.	11, 11 ± 1, 15, 20	[36,121,135]
*Cerastium tomentosum* L.	Caryophyllaceae	C	Endem. Italy	6, 9, 12, 14, 15	[132,142]
*Chrysanthemum myconis* L.	Asteraceae	T	Stenomedit.	20	[130]
*Chrysocephalum apiculatum* (Labill.) Steetz	Asteraceae	H	Endem. Italy	19	[77]
*Cistus salviifolius* L.	Cistaceae	NP	Stenomedit.	10, 13	[87]
*Clinopodium acinos* Kuntze	Lamiaceae	T	Eurimedit.	5, 10	[37]
*Colchicum autumnale* L.	Colchicaceae	G	C-Europ.	20	[130]
*Consolida regalis* Gray	Ranunculaceae	T	Eurimedit.	20	[130]
*Convolvulus cneorum* L.	Convolvulaceae	C	N-Medit.	7.5, 10, 15	[19,138,144]
*Convolvulus sabatius* Viv.	Convolvulaceae	G	W.Stenomedit.	19	[77]
*Cotinus coggygria* Scop.	Anacardiaceae	NP	Medit.-Turan.	10, 13	[87,126]
*Crithmum maritimum* L.	Apiaceae	C	Eurimedit.	7.5, 15, 20	[36,120,141,145,146]
*Crocus vernus* (L.) Hill	Iridaceae	G	Eurimedit.	20	[130]
*Delosperma cooperi* (Hook.f.) L.Bolus	Aizoaceae	C	S-Africa	14	[132]
*Delosperma* N.E.Br. ‘Kelaidis’	Aizoaceae	C	S-Africa	14	[132]
*Delosperma* N.E.Br. sp.	Aizoaceae	C	S-Africa	14	[132]
*Dianella caerulea* Sims ‘Breeze’	Asphodelaceae	H	Australia	10	[143]
*Dianthus carthusianorum* L.	Caryophyllaceae	H	C-Europ.	15, 20	[36,130]
*Dianthus deltoides* L.	Caryophyllaceae	H	Euroasiat.	10	[131]
*Dianthus deltoides* L. ‘Leuchtfunk’	Caryophyllaceae	H	Euroasiat.	14	[132]
*Dianthus fruticosus* subsp. *fruticosus*	Caryophyllaceae	H	Endem. Greece	7.5, 15	[147]
*Dianthus gratianopolitanus* Vill.	Caryophyllaceae	H	C-Europ.	6, 9, 12, 15	[142]
*Dianthus superbus* L.	Caryophyllaceae	H	Euroasiat.	5, 10	[37]
*Dorycnium hirsutum* (L.) Ser.	Fabaceae	C	Eurimedit.	14	[132]
*Drosanthemum floribundum* (Haw.) Schwantes	Aizoaceae	C	S-Africa	5, 6, 8, 10, 11 ± 1, 12	[133,135,148]
*Dymondia margaretae* Compton	Asteraceae	H	S-Africa	11 ± 1	[135,149]
*Echium plantagineum* L.	Boraginaceae	H	Eurimedit.	8	[150]
*Echium vulgare* L.	Boraginaceae	H	Europ.	8	[150]
*Emerus major* Mill.	Fabaceae	NP	C-Europ.	10, 13	[87]
*Erodium cicutarium* (L.) L’Hér.	Geraniaceae	T	Subcosmop.	10	[131]
*Erophila verna* (L.) DC.	Brassicaceae	T	Circumbor.	5, 10	[37]
*Euphorbia characias* L.	Euphorbiaceae	NP	Stenomedit.	15, 20	[36]
*Euphorbia cyparissias* L.	Euphorbiaceae	H	C-Europ.	5, 10	[37]
*Euphorbia pithyusa* L.	Euphorbiaceae	C	W-Medit.	15, 20	[36]
*Festuca arundinacea* Schreb.	Poaceae	H	Paleotemp.	8, 16	[54]
*Frankenia laevis* L.	Frankeniaceae	C	Stenomedit.	11 ± 1	[135,149]
*Geranium molle* L.	Geraniaceae	H	Eurasiat.	10	[131]
*Glaucium flavum* Crantz	Papaveraceae	H	Eurimedit.	15, 20	[36]
*Halimione portulacoides* (L.) Aellen	Amaranthaceae	C	Circumbor.	5, 10	[134,136]
*Hardenbergia violacea* (Schneev.) Stearn	Fabaceae	P	Australia	19	[77]
*Helianthemum* Gray ‘Fire Dragon’	Cistaceae	C	-	14	[132]
*Helianthemum nummularium* Mill.	Cistaceae	C	Europ.-Caucas.	5, 10	[37]
*Helichrysum italicum* (Roth) G.Don	Asteraceae	C	S-Europ.	7.5, 10, 15, 20	[19,36,86,151]
*Helichrysum italicum* subsp. *microphyllum* (Willd.) Nyman	Asteraceae	C	Eurimedit.	15, 20	[36]
*Helichrysum orientale* (L.) Gaertn.	Asteraceae	C	Endem. Medit.	7.5, 8, 10, 15	[19,86,140]
*Helichrysum stoechas* (L.) Moench	Asteraceae	C	Stenomedit.	11 ± 1, 15, 20	[36,135]
*Hemerocallis* L. ‘Stella de Oro’	Asphodelaceae	G	-	14	[132]
*Heuchera* L. ‘Electra’	Saxifragaceae	H	-	10	[152]
*Heuchera* L. ‘Obsidian’	Saxifragaceae	H	-	10	[152]
*Hypericum calycinum* L.	Hypericaceae	C	Medit.-Mont.	14	[132]
*Hypochaeris radicata* L.	Asteraceae	H	Europ.-Caucas.	10	[131]
*Hyssopus officinalis* L.	Lamiaceae	C	Orof. Eurasiat.	19	[77]
*Hyssopus officinalis* subsp. *aristatus* (Godr.) Nyman	Lamiaceae	C	Medit.	14	[132]
*Indigofera australis* Willd.	Fabaceae	P	Australia	19	[77]
*Iris chamaeiris* Bertol.	Iridaceae	G	NW-Stenomedit.	20	[130]
*Iris lutescens* Lam.	Iridaceae	G	NW-Stenomedit.	5, 10, 11 ± 1	[37,135,149]
*Jacobaea maritima* (L.) Pelser & Meijden	Asteraceae	C	Stenomedit.	19	[77]
*Lagurus ovatus* L.	Poaceae	T	Eurimedit.	5, 10	[37,153]
*Lampranthus spectabilis* (Haw.) N.E.Br.	Aizoaceae	C	S-Africa	6, 8, 10, 12	[148]
*Lavandula angustifolia* Mill.	Lamiaceae	NP	Stenomedit.	6, 8, 10, 12	[148]
*Lavandula dentata* L.	Lamiaceae	NP	Paleosubtrop.	15	[152]
*Lavandula stoechas* L.	Lamiaceae	NP	Stenomedit.	15, 20	[36]
*Lavandula stoechas* subsp. *luisieri* (Rozeira) Rozeira	Lamiaceae	NP	Endem. Spain	15	[42]
*Leontodon tuberosus* L.	Asteraceae	H	Stenomedit.	15, 20	[36,130]
*Ligustrum vulgare* L.	Oleaceae	NP	Eurasiat.	10 or 13	[87]
*Limonium virgatum* (Willd.) Fourr.	Plumbaginaceae	C	Eurimedit.	11 ± 1, 15	[120,135,141]
*Linum bienne* Mill.	Linaceae	H	Eurimedit.	5, 10	[37]
*Lobularia maritima* (L.) Desv.	Fabaceae	C	Stenomedit.	5, 10	[37,131]
*Lomandra longifolia* Labill. ‘Tanika’	Asparagaceae	H	Australia	10	[143]
*Lomelosia cretica* (L.) Greuter & Burdet	Caprifoliaceae	C	Stenomedit.	7.5, 10, 15	[19,138]
*Lotus creticus* L.	Fabaceae	C	Stenomedit.	10, 11 ± 1	[135,153]
*Medicago arborea* L.	Fabaceae	P	NE-Medit.	6, 8, 10, 12	[148]
*Melissa officinalis* L.	Lamiaceae	H	Eurimedit.	8	[140]
*Muscari comosum* (L.) Mill.	Asparagaceae	G	Eurimedit.	10	[131]
*Myoporum parvifolium* R.Br.	Scrophulariaceae	C	Australia	10	[143]
*Narcissus tazetta* L.	Amaryllidaceae	G	Stenomedit.	20	[130]
*Nepeta cataria* L.	Lamiaceae	H	E-Medit.	19	[77]
*Nigella damascena* L.	Ranunculaceae	T	Eurimedit.	20	[130]
*Olearia axillaris* (DC.) Benth.	Asteraceae	T	Australia	19	[77]
*Origanum dictamnus* L.	Lamiaceae	H	Endem. Crete	7.5, 10, 15	[19,138]
*Origanum majorana* L.	Lamiaceae	H	Saharo-Sind.	7.5, 10 or 15	[19]
*Origanum onites* L.	Lamiaceae	C	E-Medit.	8 or 16	[54]
*Origanum vulgare* L.	Lamiaceae	H	Eurasiat.	19	[77]
*Ornithogalum umbellatum* L.	Liliaceae	G	Eurimedit.	10, 20	[130,131]
*Otanthus maritimus* (L.) Hoffmanns. & Link	Asteraceae	C	Medit.-Atlant.	10, 20	[36]
*Paliurus spina-christi* Mill.	Rhamnaceae	P	SE-Europ.	10, 13	[87]
*Pallenis maritima* (L.) Greuter	Asteraceae	H	W-Medit.	7.5, 10, 11 ± 1, 15	[19,135,141,154,155]
*Pennisetum clandestinum* Hochst. ex Chiov.	Poaceae	H	E-Africa	5	[134]
*Petrorhagia prolifera* (L.) P.W.Ball & Heywood	Caryophyllaceae	T	Eurimedit.	5 and 10	[37]
*Petrorhagia saxifraga* Link	Caryophyllaceae	H	Eurimedit.	10	[131]
*Phillyrea angustifolia* L.	Oleaceae	P	Stenomedit.	10, 13	[87]
*Phlox douglasii* Hook. ‘McDaniels Cushion’	Polemoniaceae	H	N-America	14	[132]
*Pistacia lentiscus* L.	Anacardiaceae	P	S-Medit.	10, 13	[87]
*Plantago afra* L.	Plantaginaceae	T	Stenomedit.	5, 10	[37]
*Prunus mahaleb* L.	Rosaceae	P	S-Europ.	10, 13	[87,126]
*Pyrus pyraster* (L.) Burgsd.	Rosaceae	P	Eurasiat.	10, 13	[87]
*Rosmarinus officinalis* L.	Lamiaceae	NP	Stenomedit.	8, 15, 19	[42,77,140]
*Salvia fruticosa* Mill.	Lamiaceae	C	E-Medit.	8, 10	[123,140,156]
*Salvia* L. hybr	Lamiaceae	C	-	10	[123,156]
*Salvia officinalis* L.	Lamiaceae	C	Stenomedit.	10, 13, 18	[63,87,123,156]
*Salvia officinalis* L. ‘Berggarten’	Lamiaceae	C	Stenomedit.	10	[152]
*Salvia pomifera* subsp. *pomifera*	Lamiaceae	C	Endem. Crete	10	[123]
*Salvia ringens* Sm.	Lamiaceae	C	Endem. Greece	10	[123,156]
*Salvia tomentosa* Mill.	Lamiaceae	C	NE-Medit.	10	[123]
*Santolina chamaecyparissus* L.	Asteraceae	NP	Medit.	7.5, 10, 15	[19]
*Santolina rosmarinifolia* L.	Asteraceae	NP	Medit.	11 ± 1	[135]
*Saponaria ocymoides* L.	Caryophyllaceae	H	Orof. S-Europ.	14	[132]
*Satureja illyrica* Host	Lamiaceae	C	S-Europ.	14	[132]
*Satureja montana* L.	Lamiaceae	C	W-Medit.	15, 20	[36,151]
*Scabiosa columbaria* L.	Dipsacaceae	H	Eurasiat.	15, 20	[36,130]
*Scilla autumnalis* L.	Asparagaceae	G	Eurimedit.	20	[130]
*Scrophularia canina* L.	Scrophulariaceae	H	Eurimedit.	15, 20	[36]
*Scrophularia peregrina* L.	Scrophulariaceae	T	Stenomedit.	10	[131]
*Sedum acre* L.	Crassulaceae	C	Europ.	4, 5, 6, 7, 10, 12	[37,128,131,157]
*Sedum album* L.	Crassulaceae	C	Eurimedit.	4, 5, 6, 7, 10, 12, 14	[37,128,131,132,133,157,158]
*Sedum floriferum* Praeger	Crassulaceae	C	Siberia	14	[132]
*Sedum hispanicum* L.	Crassulaceae	C	SE-Europ.	5, 14	[132,133]
*Sedum* L. spp.	Crassulaceae	C	-	6, 10	[129,152]
*Sedum ochroleucum* Chaix	Crassulaceae	C	Medit.-Mont.	5	[133]
*Sedum reflexum* L.	Crassulaceae	C	C-Europ.	6, 12, 14	[132,157]
*Sedum rupestre* L.	Crassulaceae	C	C-Europ.	15, 20	[36]
*Sedum sediforme* (Jacq.) Pau	Crassulaceae	C	Stenomedit.	5, 6, 7.5, 8, 10, 11, 12, 13, 15, 16, 19	[54,65,120,121,133,141,148,158,159,160]
*Sedum sexangulare* L.	Crassulaceae	C	C-Europ.	6, 10, 12, 14	[132,157,158]
*Sedum spurium* M.Bieb.	Crassulaceae	C	Europ.-Caucas.	14	[132]
*Sedum spurium* M.Bieb. cf. ^6^ ‘Coccineum’	Crassulaceae	C	Europ.-Caucas.	10	[158]
*Sedum spurium* M.Bieb. cf. ^6^ ‘Summer Glory’	Crassulaceae	C	Europ.-Caucas.	10	[158]
*Sempervivum* L. ‘Reinhard’	Crassulaceae	C	-	10	[152]
*Sesuvium verrucosum* Raf.	Aizoaceae	C	America	5	[134]
*Sideritis athoa* Papan. & Kokkini	Lamiaceae	C	Macarones.	7.5, 10, 15	[19,138]
*Sideritis hyssopifolia* L.	Lamiaceae	C	NW-Medit.	5, 10	[37]
*Silene conica* L.	Caryophyllaceae	T	Paleotemp.	5, 10	[37]
*Silene gallica* L.	Caryophyllaceae	T	Eurimedit.	10	[131]
*Silene vulgaris* (Moench) Garcke	Caryophyllaceae	H	Paleotemp.	5, 10	[153]
*Spartium junceum* L.	Fabaceae	P	Eurimedit.	10, 13	[87]
*Sporobolus pungens* Kunth	Poaceae	G	Subtrop.	15	[120,141]
*Stachys byzantina* K.Koch	Lamiaceae	H	E-Asia	10, 14	[132,152]
*Sternbergia lutea* (L.) Ker Gawl. ex Spreng.	Amaryllidaceae	G	Medit.-Mont.	20	[130]
*Teucrium chamaedrys* L.	Lamiaceae	C	Eurimedit.	14	[132]
*Teucrium fruticans* L.	Lamiaceae	NP	Stenomedit.	19	[77]
*Thymus caespititius* Brot.	Lamiaceae	C	Iberian Peninsula	15	[151]
*Thymus marschallianus* Willd.	Lamiaceae	C	Eurasiat.	14	[132]
*Thymus pseudolanuginosus* Ronniger	Lamiaceae	C	S-Europ.	15	[151]
*Thymus serpyllum* L.	Lamiaceae	H	Eurasiat.	11 ± 1	[135]
*Trifolium arvense* L.	Fabaceae	T	Paleotemp.	10	[131]
*Trifolium campestre* Schreb.	Fabaceae	T	Paleotemp.	10	[131]
*Tuberaria guttata* (L.) Fourr.	Cistaceae	T	Eurimedit.	20	[130]
*Verbascum blattaria* L.	Scrophulariaceae	H	Cosmopol.	10	[131]
*Verbascum thapsus* L.	Scrophulariaceae	H	Europ.-Caucas.	15, 20	[36]
*Veronica prostrata* L.	Plantaginaceae	H	Eurasiat.	14	[132]
*Zoysia matrella* (L.) Merr.	Poaceae	H	E-Asia	7.5, 15	[161]

^1^ We considered that substrate layer thickness of extensive green roof was from 5 to 20 cm [158]; ^2^ the names of the species have been corrected based on the indications of “World flora online” [162]; we preferred to keep the names indicated by the authors of the papers, except when multiple synonyms of the same species were used; in this case, we have only reported the correct botanical name; ^3^ according to “World flora online” [162]; ^4^ according to The Life Forms of Plants of Raunkiaer [163]; H = Hemicryptophyte, NP = Nanophanerophyte, G = Geophyte, T = Therophyte, C = Chamaephyte, P = Phanerophyte; ^5^ references are mainly related to Pignatti et al. [164]; ^6^ “cf.” means that, according to the authors, the cultivar cannot be affirmed with certainty but the phenotype is compatible.

## Data Availability

Not applicable.

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
