# Peer review of "Which Plant Species for Green Roofs in the Mediterranean Environment?"

_plants, 2023, doi:10.3390/plants12233985_

Round 1
Reviewer 1 Report
Comments and Suggestions for Authors
Dear Authors,
this is interesting and well written approach to the problem of widening our knowledge of green spaces in cities discussing anthropogenic and technogenic impact on plants within so called ‘green roofs’.
The paper is well structured, Authors applied proper methodology. However it is missing some necessary aspects related to soils of green roofs, e.g. designing to sustaint life as it plays major role in supporting plant communities, what is main paper subject, also literature review shall be subsequently enhanced in this matter. I would then recommend considering some more aspects, which should be discussed as paragraph before or after “2. The environmental conditions of green roofs in a Mediterranean environment” ; some suggestion for literature:
· Soil formation on green roofs and its contribution to urban biodiversity with emphasis on Collembolans by Stefan Schrade, Matthias Böning;
· Soil-Based Green Roofs by Brooke Byerley Best, Rebecca K. Swadek and Tony L. Burgess.
·
Also I would like to suggest considering edifisols in overview, as naturally creating “green roofs” due to colonization of ruderal plants, it can be interesting insight into what species can survive and develop in roof environment; see:
· Edifisols—a new soil unit of technogenic soils by Charzyński et al. (Journal of Soils and Sediments);
· Characteristics of soil organic matter of edifisols – An example of technohumus system by M. Markiewicz et al (Applied Soil Ecology).
sincerely yours,
Reviewer
Comments on the Quality of English Languageany
Author Response
Dear reviewer,
The authors would like to thank you for your comments. The manuscript has been accordingly revised. Corrections and suggestions have been implemented in the current version of the manuscript. All the modifications are highlighted in yellow in the manuscript. We hereby provide a point-by-point answer.
The authors
Dear Authors,
this is interesting and well written approach to the problem of widening our knowledge of green spaces in cities discussing anthropogenic and technogenic impact on plants within so called ‘green roofs’.
The paper is well structured, Authors applied proper methodology. However it is missing some necessary aspects related to soils of green roofs, e.g. designing to sustaint life as it plays major role in supporting plant communities, what is main paper subject, also literature review shall be subsequently enhanced in this matter. I would then recommend considering some more aspects, which should be discussed as paragraph before or after “2. The environmental conditions of green roofs in a Mediterranean environment” ; some suggestion for literature:
- Soil formation on green roofs and its contribution to urban biodiversity with emphasis on Collembolans by Stefan Schrade, Matthias Böning;
- Soil-Based Green Roofs by Brooke Byerley Best, Rebecca K. Swadek and Tony L. Burgess.
Also I would like to suggest considering edifisols in overview, as naturally creating “green roofs” due to colonization of ruderal plants, it can be interesting insight into what species can survive and develop in roof environment; see:
Edifisols—a new soil unit of technogenic soils by Charzyński et al. (Journal of Soils and Sediments);
Characteristics of soil organic matter of edifisols – An example of technohumus system by M. Markiewicz et al (Applied Soil Ecology).
sincerely yours,
Reviewer
Author Answer (A.A.): thanks for the comments and suggestions about the substrates and their influence on plant performance. The paper and some comments are added to the manuscript.

Reviewer 2 Report
Comments and Suggestions for Authors
The paper is a critical analysis of references on green roofs to identify suitable criteria for plant species selection. It is in general well-written and clear, and it also deals with an interesting topic that will increase in the future. In particular, the discussion on Criteria for plant species selection such as the idiotype of plant species is interesting and clearly explained. Its publication is welcomed, since it gives several elements of analysis for planting species on green roofs.
In my opinion the weak point is Table 2, which reports information on the plant species proposed for extensive green roofs in the Mediterranean region. Such Table should be improved, adding other important information, such as:
- Data set of information don’t derive by a comprehensive literature review, but from some relevant papers. The A. should clarify the origin of the data set.
- Performance of species in the experiments done (the presence in the tests don’t means that the species is suitable). The A. should add a column on this and synthetize the results.
- The chorotype of the species is also relevant, to analyze if the plant is native in the Mediterranean region or not.
Furthermore, the possible use of shrubs, such as Arbutus unedo L Cotinus coggygria Scop. and Prunus mahaleb and many others..... is surprising in EGR, due to the limited soil depth. Please give some comment on this, because often the paper that suggest such possibility only consider the first stages of growth, and don’t analyze the behavior in a longer time.
Minor remarks
In the first paragraph (par. 2)., some considerations are very general and not limited to the Mediterranean environment. So, I suggest remove from the title the geographical context.
123-124 It has been observed that positive interactions can be determined between plant species... such phrase is very general... please give more comments, since also the contrary is possible.
I suggest reviewing one point in Table 1. Key traits, because I believe it is not a good idea plant here rare and endangered plant species.
Author Response
Dear reviewer,
The authors would like to thank you for your comments. The manuscript has been accordingly revised. Corrections and suggestions have been implemented in the current version of the manuscript. All the modifications are highlighted in yellow in the manuscript. We hereby provide a point-by-point answer.
The authors
The paper is a critical analysis of references on green roofs to identify suitable criteria for plant species selection. It is in general well-written and clear, and it also deals with an interesting topic that will increase in the future. In particular, the discussion on Criteria for plant species selection such as the idiotype of plant species is interesting and clearly explained. Its publication is welcomed, since it gives several elements of analysis for planting species on green roofs.
Author Answer (A.A.): Thanks for your positive comments and suggestions.
In my opinion the weak point is Table 2, which reports information on the plant species proposed for extensive green roofs in the Mediterranean region. Such Table should be improved, adding other important information, such as:
- Data set of information don’t derive by a comprehensive literature review, but from some relevant papers. The A. should clarify the origin of the data set.
- Performance of species in the experiments done (the presence in the tests don’t means that the species is suitable). The A. should add a column on this and synthetize the results.
- The chorotype of the species is also relevant, to analyze if the plant is native in the Mediterranean region or not.
A.A.: Your comments have been very helpful and inspiring. Regarding the origin of the data and, therefore, the identification of the articles, we selected studies that analyzed, through experimental trials carried out in a Mediterranean environment, the performance of different plant species on extensive roofs. For the latter, as reported in the legend, we adopted the definition reported by Perez et al. (2020); therefore, green roofs with substrate depths of up to 20 cm were used. Reporting the performance of different species would certainly have represented important information, but the heterogeneity of the operative conditions and objectives would have led, in our opinion, to misleading information. The different species, in fact, were often subjected to different types of stress (water, high, and low temperatures), so performance was a function of these suboptimal conditions. Reporting that a plant species was suitable, perhaps because the goal was to study roof runoff or energetic performances, could become misleading, because the species was not subjected to suboptimal conditions in other trials. The same plant species may not be able to resist one of the stresses analyzed by other authors. The list proposed in Table 2 is of interest to understand the type of plant; for this reason, we have reported Raunkier's plant life forms and fully accepted your suggestion, and we have now added the chorotype of the plants. The objective of this study was to demonstrate the biological diversity of this species, which, in the Mediterranean environment, has been evaluated in extensive green roofs.
Furthermore, the possible use of shrubs, such as Arbutus unedo L Cotinus coggygria Scop. and Prunus mahaleb and many others ..... is surprising in EGR, due to the limited soil depth. Please give some comment on this, because often the paper that suggest such possibility only consider the first stages of growth, and don’t analyze the behavior in a longer time.
A.A.: Thank you for your comments which help us to better explain our idea. In the papers we have analysed, relating to the use of shrubs in extensive green roofs, are adopted plants and not rooted cuttings. For example, Savi and collaborators (2015 and 2016) analyzed 2–3-year-old shrubs previously grown in small pots. The authors are convinced that this is a paradox, as demonstrated by the title of one of the works: “Does shallow substrate improve water status of plants growing on green roofs? Testing the paradox in two sub-Mediterranean shrubs” (Savi et al., 2015). There are no indications on the age of the Arbutus unedo specimens used in the trial by Raimondo et al. (2015) but the initial height was 43 cm and therefore they were not in the first stages of growth. The results of these trials have highlighted how the reduced depth of the substrate translates into less severe water stress than hypothesized and that the lower depth of the substrate indirectly stimulates lower water consumption as a consequence of the reduced plant biomass, so it is possible to hypothesize a green roof with the use of shrubs resistant to stress in sub-Mediterranean areas even in the presence of a substrate only 10 cm deep. Comments on these aspects is now present in the text.
Minor remarks
In the first paragraph (par. 2)., some considerations are very general and not limited to the Mediterranean environment. So, I suggest remove from the title the geographical context.
A.A.: Thank for your suggestion; we eliminated in the paragraph 2 the reference to the Mediterranean environment.
123-124 It has been observed that positive interactions can be determined between plant species... such phrase is very general... please give more comments, since also the contrary is possible.
A.A.: The sentence was modified according your suggestion.
I suggest reviewing one point in Table 1. Key traits, because I believe it is not a good idea plant here rare and endangered plant species.
A.A.: Thank you for the suggestion. From an ecological point of view, it is desirable that green roof could be a provision of habitat for rare and endangered species, but the table is entitled to key traits for suitable species for green roof; for this reason, the sentence is not correct. We eliminated it.
